# Inbox - a messaging app based on Linked Data Notifications and Solid

Antonín Karola and Jakub Klímek[0000−0001−7234−3051]

Department of Software Engineering, Faculty of Information Technology,
Czech Technical University in Prague, Thákurova 9, 160 00 Praha 6, Czech Republic
karolan1@fit.cvut.cz, klimejak@fit.cvut.cz

**Abstract.** In the context of the recent web re-decentralization and true data ownership initiative, several W3C Recommendations building on top of linked data were published. These include the Linked Data Platform for basic read-write capabilities, Linked Data Notifications (LDN) as a simple messaging protocol, and Activity Streams 2.0 (AS2) as a message content vocabulary. Together with WebID for user identification, they are used within the Solid ecosystem by various applications. However, the majority of current Solid-based applications are technological demonstrators, aiming mainly at developers rather than end-users. In this demonstration, we present Inbox, a Solid, LDN, and AS2 based messaging application. It aims at being a user-friendly demonstrator of the recent technologies, resembling an ordinary e-mail client with the expected functionalities. These include reading messages from inboxes, getting notified of an incoming message, and being able to send messages to arbitrary inboxes and to user's contacts present in their Solid Pod.

**Keywords:** Solid · Linked Data Notifications · ActivityStreams · messaging

## 1 Introduction

The Web was designed as a decentralized network from its start. However, market monopolization has introduced a new problem - web centralization. The tech giants have made their users dependent on them for information and data storage and compete over the ownership of their user's data. Consequently, web applications of different vendors are incapable of intercommunication. For instance, a Facebook user cannot comment on YouTube and, conversely, YouTube cannot send you notifications to the app of your choice. This is known as the problem of data silos. As we can see in the recent US events[1], based on the collected user data, these companies can influence politics, access private messages, and delete user content without any justification.

Recently, these problems became the focus of the initiative around the Solid project[2], supporting Web re-decentralization and regaining of control over user's

---

[1] https://www.theguardian.com/technology/commentisfree/2019/oct/23/facebook-influence-next-election-democratic

[2] https://solidproject.org/

own data. The W3C has published technical recommendations supporting the initiative, namely the Linked Data Platform 1.0 [4] for basic linked data read and write capabilities, Linked Data Notifications (LDN) [1] as a simple messaging protocol and Activity Streams 2.0 (AS2) [3] as a message content vocabulary. However, the majority of current applications implementing these technologies are aimed at developers and enthusiasts, showing them how the technologies can be used by them in their applications. For end-users, the ecosystem still remains too immature, exotic, inaccessible, and hard to use.

In this demonstration, we present Inbox - an end-user-focused web application for messaging based on Solid. Its main aim is to show how modern, decentralized Web technologies can be used in everyday life, bringing them closer to the end-users and promoting the adoption of Solid Pods.

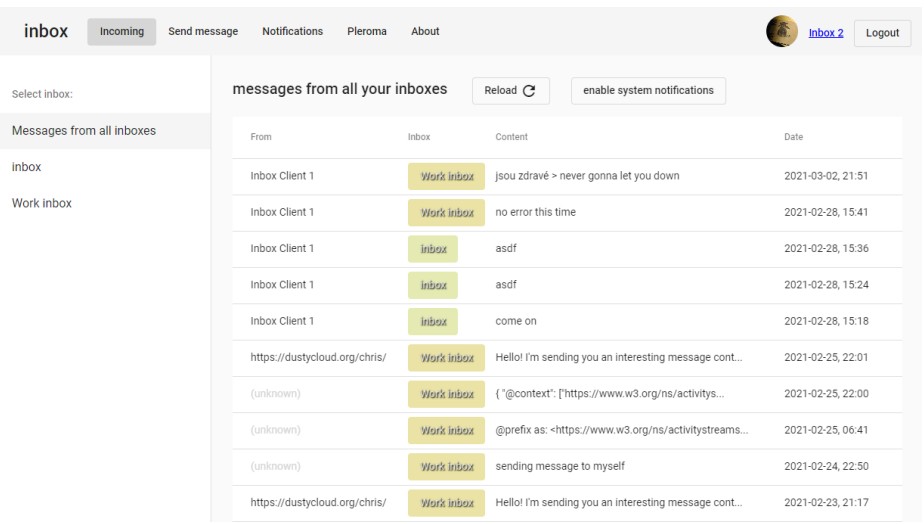

**Fig. 1.** List of messages from various monitored LDN inboxes

## 2 Inbox and Demonstration

Inbox aims to provide a real-world example of a messaging application built using Solid and the recent W3C Recommendations while offering a familiar, e-mail like functionality. Inbox allows the users to login using their WebID[3], monitor their LDN inbox, read their messages (a.k.a. LDN notifications), send messages to their contacts, and get notified when a new message arrives. Multiple inboxes can be monitored, starting with the one associated with the user's WebID hosted in their Solid Pod, and also including other LDN inboxes, added either using their

---

[3] https://www.w3.org/2005/Incubator/webid/spec/

IRIs or the IRIs of the LDN Resources having those LDN inboxes. Inbox is built using the Angular framework[4]. To work with Solid Pods, the application is using the official Solid client libraries[5], namely *solid-client-authn-js*[6] for user authentication and *solid-client-js*[7] to access and data from Solid Pod. The source code is hosted on GitHub `https://github.com/WhyINeedToFillUsername/inbox`.

The demonstration will be performed on our live demo instance running on `https://whyineedtofillusername.github.io/inbox/`. We will go through the following functionality of Inbox:

*1. WebID login* The user needs to login using their WebID first, to be able to monitor their inboxes. Also, the data of Inbox itself including the list of monitored inboxes is stored in the user's Solid Pod.

*2. Start monitoring an arbitrary inbox* The user can add an arbitrary LDN inbox to their list of monitored inboxes and get notified when a message arrives in that inbox. By default, the user sees the content of their default LDN inbox linked from their WebID profile in their Solid Pod.

*3. List messages in a selected inbox* The user can access the list of messages in a selected inbox.

*4. List messages in all monitored inboxes* The user can access the list of messages in all monitored inboxes (see Figure 1).

*5. Read message content* The user can select a message from the message list and can see the message content. If the message contains an AS2 message serialized in JSON-LD [2], it gets parsed into a user-friendly view (see Figure 2).

*6. Reply to message* Given the incoming message was parsed and contained sender information, the user can reply to the selected message.

*7. Send new message* The user can write a new message. The recipients can be selected from the list of the user's contacts, stored in their profile in their Solid Pod using the `foaf:knows` property. Alternatively, the recipients can be inserted directly, either using their WebID or using the IRI of the LDN inbox.

*8. Get notified when a message is received* The user can be notified using the operating system's notification when a new message arrives. This functionality uses the Notifications API[8].

---

[4] `https://angular.io/`

[5] `https://docs.inrupt.com/developer-tools/javascript/client-libraries/`

[6] `https://github.com/inrupt/solid-client-authn-js`

[7] `https://github.com/inrupt/solid-client-js`

[8] `https://developer.mozilla.org/en-US/docs/Web/API/notification`

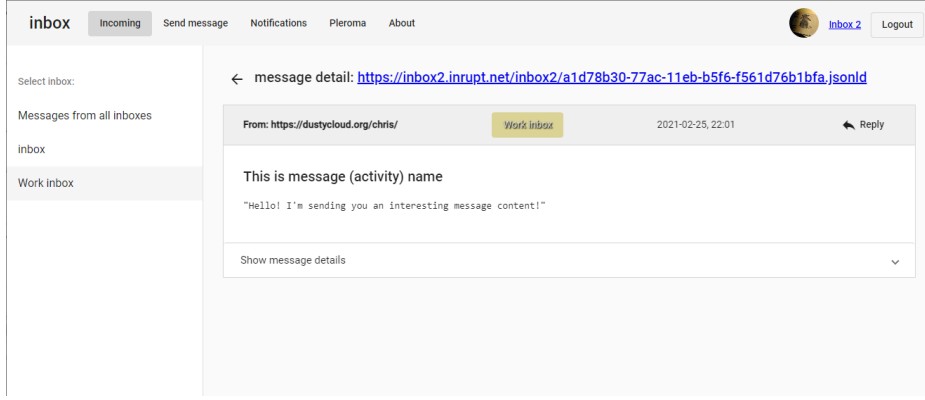

**Fig. 2.** Example of a parsed received message with Activity Streams 2.0 content

## 3 Conclusions and Future Work

This demonstration shows how the Linked Data Notifications recommendation can be used today by end-users via a user-friendly messaging application. Doing so also promotes the usage of Solid Pods and WebIDs for authentication, authorization, and storage of user data.

During the development, we ran into issues caused by the Solid related libraries still being under development. For instance, the authorization library does not keep the login state after redirect[9]. This means that after hitting page refresh in their browser, the user has to login again. Moreover, using the community Solid Pod for data storage causes some performance issues. Without any caching strategy, the Inbox application is currently producing many HTTP requests to the Solid server. Therefore, future work can introduce a caching intermediary in order to reduce the number of requests and improve the user experience. Also, features such as pagination of the message lists and an option to delete or archive old messages could be introduced.

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
