# OpenReview forum: "Inbox - a messaging app based on Linked Data Notifications and Solid"
_eswc-conferences.org/ESWC/2021/Conference/Poster_and_Demo_Track — Submitted to ESWC2021 P&D_

### Official Review · AnonReviewer2 · 2021-04-15
**A user interface for messaging application**

**Rating:** 3
**Confidence:** 4

**Review:**

This paper describes a user interface for a messaging application using Linked Data Notification, Solid Pod, and WebID. However, the research/resource/in-use contribution is not presented in any way. It may not suitable for a conference like ESWC.

**Anonymity:**

Yes, I would like my review to remain anonymous.

---

### Official Review · AnonReviewer1 · 2021-04-16
**Interesting paper of a Solid-based application**

**Rating:** 7
**Confidence:** 4

**Review:**

This paper presents a description of Inbox: a Solid-based messaging application focused on end-user usability. It describes different features which are aligned with what is expected from these kind of applications.

In my opinion these kind of papers are long overdue from the Solid community side. It is important to show the feasibility of the approach across different use cases and highlight the technical challenges that need to be addressed both by the academic community and the industry, if they want to realize their vision. In that sense, I think this paper takes a good first step in the right direction.

However, I would recommend the authors to briefly position their approach (i.e. Solid) with respect to other alternatives out there. How does this approach compares, for example to Mastodon. Are there any others?
Also, when you describe the different features of Inbox, add a brief description of what is happening behind the scenes. What kind of interactions occur between the application and the Solid pods when you click, for example, on login or send message?


**Anonymity:**

Yes, I would like my review to remain anonymous.

---

### Official Review · AnonReviewer4 · 2021-04-16
**Interesting idea and practical usefulness, but no research value**

**Rating:** 2
**Confidence:** 4

**Review:**

This paper suggests a linked data empowered messaging app using Solid. The following is a list of needed improvements to the paper:

- some typos need to be corrected
- missing a baseline and research question
- implementation details are missing, there is only a list of functionality and a link to the github page, no experiments
- there is no results and discussion
- the live demo is accessible but needs a login which is not provided in the paper (reviewers should not be required to create an account).

Overall while the idea sounds interesting, this is not a research paper, rather just an info sheet for the implementation of an app.

**Anonymity:**

Yes, I would like my review to remain anonymous.

---

### Decision · Program_Chairs · 2021-04-19

Reject